

# Tree regeneration and ontogenetic strategies of northern European hemiboreal forests: transitioning towards closer-to-nature forest management

Raimundas Petrokas[1], Michael Manton[2] and Darius Kavaliauskas[1,2]

[1] Department of Forest Genetics and Tree Breeding, Institute of Forestry, Lithuanian Research Centre for Agriculture and Forestry, Girionys, Kaunas, Lithuania
[2] Bioeconomy Research Institute, Vytautas Magnus University, Akademija, Kaunas, Lithuania

Corresponding author
Raimundas Petrokas,
raimundas.petrokas@lammc.lt

## ABSTRACT

**Background:** Tree ontogeny is the genetic trajectories of regenerative processes in trees, repeating in time and space, including both development and reproduction. Understanding the principles of tree ontogeny is a key priority in emulating natural ecological patterns and processes that fall within the calls for closer-to-nature forest management. By recognizing and respecting the growth and development of individual trees and forest stands, forest managers can implement strategies that align with the inherent dynamics of forest ecosystem. Therefore, this study aims to determine the ontogenetic characteristics of tree regeneration and growth in northern European hemiboreal forests.

**Methodology:** We applied a three-step process to review i) the ontogenetic characteristics of forest trees, ii) ontogenetic strategies of trees for stand-forming species, and iii) summarise the review findings of points i and ii to propose a conceptual framework for transitioning towards closer-to-nature management of hemiboreal forest trees. To achieve this, we applied the super-organism approach to forest development as a holistic progression towards the establishment of natural stand forming ecosystems.

**Results:** The review showed multiple aspects; first, there are unique growth and development characteristics of individual trees at the pre-generative and generative stages of ontogenesis under full and minimal light conditions. Second, there are four main modes of tree establishment, growth and development related to the light requirements of trees; they were described as ontogenetic strategies of stand-forming tree species: gap colonisers, gap successors, gap fillers and gap competitors. Third, the summary of our analysis of the ontogenetic characteristics of tree regeneration and growth in northern European hemiboreal forests shows that stand-forming species occupy multiple niche positions relative to forest dynamics modes.

**Conclusions:** This study demonstrates the importance of understanding tree ontogeny under the pretext of closer-to-nature forest management, and its potential towards formulating sustainable forest management that emulates the natural dynamics of forest structure. We suggest that scientists and foresters can adapt closer-to-nature management strategies, such as assisted natural regeneration of trees, to improve the vitality of tree communities and overall forest health. The presented approach prioritizes ecological integrity and forest resilience, promoting

assisted natural regeneration, and fostering adaptability and connectivity among plant populations in hemiboreal tree communities.

## INTRODUCTION

In 2021, the *European Commission (2021)* published the New EU Forest Strategy for 2030. The strategy continuously supports and enhances attempts to balance forest resource use between social, economic, and conservation aspects. It calls for the maintenance of these three interrelated aspects under the guise of closer-to-nature forest management (*European Commission, 2023*). Closer-to-nature forest management is a new approach that prioritizes the emulation of natural ecological processes and the integration of ecosystem dynamics into forest management practices. This approach contrasts traditional methods of intensive harvesting and industrial forestry models or clear-cutting by seeking a balance between human needs and the preservation of ecological integrity. The key principles that characterise closer-to-nature forest management involve selective logging and the retention of habitat structures, promoting natural regeneration, applying adaptive management, maintaining ecological connectivity, mimicking natural patterns and processes, and engaging the community and stakeholders (*Larsen et al., 2022*; *European Commission, 2023*). Despite this, commercial forestry is still considered one of the most significant achievements of scientific forest management, based on the anthropocentric paradigm of our relationship with nature (*Studley, 2010*). Policy objectives also continue to advocate for further forestry intensification as a mitigation measure of climate change (*Bäck et al., 2017*). For example, in the hemiboreal vegetation zone, an issue with finding alternatives to clear-cutting forest management practices has been an ongoing discussion (*Jõgiste et al., 2017*), precisely because there is a prevailing opinion that the main shade-tolerant tree species (which is also economically important) is Norway spruce. However, Norway spruce is associated with many problems (*i.e.*, drought, bark beetle attacks, wind throws, and root rot damages), so the implementation of alternative practices management is required but proving difficult (*de Groot, Diaci & Ogris, 2019*). This also implies that closer-to-nature forest management must be consistent with site history and consider disturbance-induced changes in biotic and abiotic factors, as the sensitivity of an ecosystem to climate change is primarily determined by its ability to recover from disturbances (*Attiwill, 1994*; *Kröel-Dulay et al., 2015*).

Focussing on emulating natural patterns and processes suggests that harnessing and assisting natural regeneration of forests should be a principal establishment method for new or rotational forest stands (*European Commission, 2023*). Natural regeneration facilitates the spontaneous growth of a variety of native forest species by improving the adaptive aptitude and resilience of forests, thus making forests more robust to adapt and

mitigate climate change (*Huuskonen et al., 2021*). The loss of genetic diversity is generally not expected in natural regeneration under current climate and local growth conditions, provided that a sufficiently large number of parental trees within the stand surrounding forests actively contribute to the reproduction process (*Konnert & Hosius, 2010*). However, limitations in genetic variability and structure may arise in natural regeneration, particularly when population sizes are dramatically reduced, resulting in reduced seed production (*Savolainen & Kärkkäinen, 1992*). Similar risks of decreasing genetic diversity (*e.g.*, effective population size) can be faced in species with low tree densities, whether due to rarity or wide dispersion, leading to less diverse parental combinations in subsequent generations (*Lande & Barrowclough, 1987*). Although the natural regeneration of forest trees has adapted to the current climate and local growth conditions, forests may not evolve in time to cope with current climate change predictions (*Aitken et al., 2008*; *Leech, Almuedo & O'Neill, 2011*; *Aitken & Bemmels, 2016*; *Valladares, 2017*). This is because trees are long-lived and limited in their ability to adapt under such expediated environmental and climate change (*Aitken et al., 2008*; *Kremer et al., 2012*; *Kijowska-Oberc et al., 2020*; *Bisbing et al., 2021*). This means that assisted species and population migration and active restoration will be needed to help increase and also to maintain forest resilience (*Richardson et al., 2009*; *Leech, Almuedo & O'Neill, 2011*; *Pedlar et al., 2012*; *Schwartz et al., 2012*; *Koralewski et al., 2015*; *Aitken & Bemmels, 2016*; *Chen et al., 2022*; *Stanturf, Ivetić & Dumroese, 2024*). Thus, closer-to-nature forest management that aims at natural forest regeneration must consider the ecophysiological characteristics of species (*i.e.*, shade tolerance, growth rate, phenology, *etc.*), which are deeply rooted in the principles of plant ontogeny.

Plant ontogeny refers to the entire sequential regenerative process of an individual plant from its initiation as a seed, through germination, seedling establishment, vegetative growth, reproductive maturity, and eventually to senescence or death (*Rolston, 2002*). According to the concept of biological age, the regenerative process of individual plants is split into several ontogenetic phases that interact uniquely with multiple environmental components (*Grubb, 1977*). These phases are integral to comprehending the complex dynamics of forest ecosystems, particularly under closer-to-nature forest management (*European Commission, 2021*). Therefore, understanding the ontogeny of trees is crucial for the implementing of closer-to-nature forest management that aligns with the inherent life cycles of trees and other vegetation in each forest ecosystem. This highlights the need for a conceptual framework to mitigate barriers to dynamic relationships among the elements that make up the forest ecological space.

The aim of the study is to characterize natural ecological patterns and processes of northern European hemiboreal forests from the point of view of the biological age dynamics of stand-forming tree species. To do this we reviewed i) the ontogenetic characteristics of forest trees, ii) ontogenetic strategies of trees for stand-forming species, and iii) summarized the findings of objectives of points i and ii to propose a conceptual framework for transitioning towards closer-to-nature management of hemiboreal forest trees.

## MATERIALS AND METHODS

### Study area and objects

The hemiboreal forest zone is the overlapping zone between the southern margin of the boreal forest zone and the northern margins of the temperate forest zone. The European hemiboreal forest extends from southern Scandinavia in the west, through the Baltic states, Russia's Kaliningrad region, southern Finland, Belarus and, and eastward to the Ural mountains in central Russia (*Ahti, Hämet-Ahti & Jalas, 1968*). We selected Lithuania's hemiboreal forest ecosystems for this review, because their natural patterns and processes are at risk due to a changing climate, forest management intensification, and planting of Norway spruce or Scots pine monocultures over the past century (*Manton et al., 2022*; *Petrokas & Manton, 2023*).

Despite the variety of climates, soils, and evolutionary backgrounds of forests in different parts of the world, the patterns of stand dynamics can be remarkably similar (*Camp & Oliver, 2004*). Forests go through a number of stages, from stand initiation, stand, self-thinning, the emergence of the underwood to its maturity. This is deeply influenced by three primary factors: climatic, edaphic, and biotic (*Bonan & Shugart, 1989*; *Larsen, 2013*). Climatic factors comprise prevailing weather conditions, including temperature, precipitation, and sunlight, which significantly impact the type of vegetation that can exist in a given forest region (*Richards, 1952*). The word 'climate' originated from the Greek word 'klima' which alludes to the slope or angle of the sun's rays descending on the Earth's surface. Edaphic factors relate to soil characteristics, such as texture, composition, moisture, and nutrient content, that shape the forest's ability to support a diverse range of plant communities and overall ecosystem health (*Pianka, 2000*). Biotic factors, involves living organisms within the forest ecosystem and encompasses the intricate interactions among various plant and animal species. These three factors contribute directly to the potential natural vegetation, habitat structures, ecological balance, and overall resilience of the forest ecosystem (*Bonan & Shugart, 1989*).

The climax, a continuum of end communities that vary in time and space across environmental gradients largely characterised by local variations in climatic, edaphic, and biotic conditions, sometimes referred to as the 'potential natural vegetation', was considered in this review as a position of relative stability characteristic of a forest ecosystem (*Richards, 1952*; *Whittaker, 1953*; *Stern & Roche, 1974*; *Kotar, 1997*). The concept of potential natural vegetation corresponds to the hypothetical state of vegetation that could arise under actual environmental conditions if human influence ceased and progressive ecological process became instantaneous (*Capelo et al., 2007*). The potential natural vegetation paradigm assumes for a given area a univocal correspondence between a single combination of bioclimatic stage and lithology, considering the biogeographical context, and a unique progressive ecological sequence leading to a single climax community. Thus, the mosaic of hemiboreal forest communities in zonal habitats is largely the result of ecological succession leading ultimately to climatic climax or potential natural vegetation. In some parts of the hemiboreal forest zone, there are areas where local peculiarities of the soil or topography make the development of the climatic climax forest

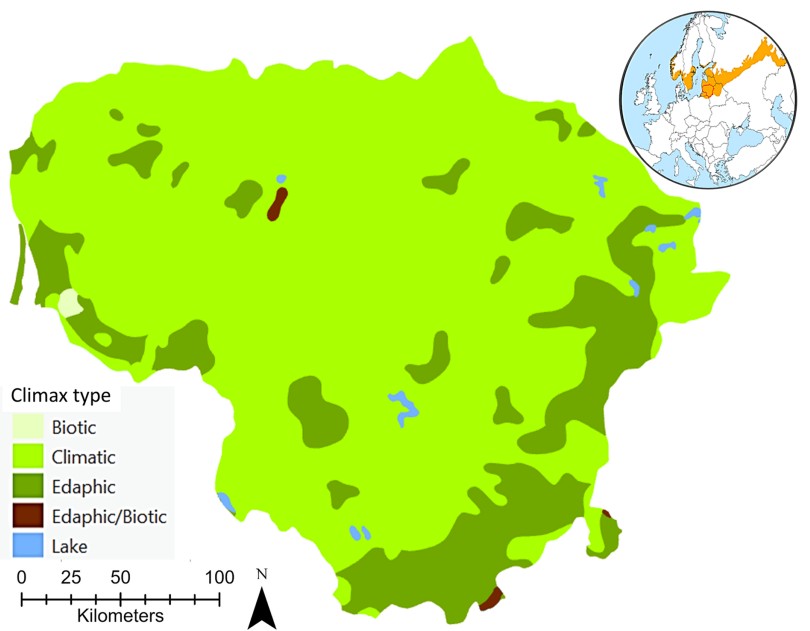

**Figure 1 An overview of the potential natural hemiboreal forest climax types of Lithuania.** Euro-VegMap 2.0.6 was used to create this map (*Bohn et al., 2004*). The global map shows both the location of Lithuania and the European hemiboreal forest region (*Ahti, Hämet-Ahti & Jalas, 1968*).

**Table 1 Potential natural forest vegetation types of Lithuania.**

| Potential natural forest vegetation types | Forest type series | Climax type |
|---|---|---|
| Hemiboreal spruce forests with broadleaved trees | oxn, ox, mox* | Climatic |
| Boreal and hemiboreal pine forests and Baltic pine groves | vm, m | Edaphic |
| Hemiboreal pine forests | v, cl | Edaphic |
| Pine bog forests | msp, csp, lsp | Biotic |
| Species-rich oak-hornbeam and pine-oak forests | hox | Climatic |
| Lime-oak forests | aeg, cmh | Climatic |
| Swamp and fen forests | fil, ur, cal/cir, c | Edaphic/Biotic |

**Note:**
*Ground layer codes of the main types of plant communities (*Karazija, 1988*): aeg—*Aegopodiosa*, c—*Caricosa*, cal—*Calamagrostidosa*, cir—*Carico-iridosa*, cl—*Cladoniosa*, cmh—*Carico-mixtoherbosa*, csp—*Carico-sphagnosa*, fil—*Filipendulo-mixtoherbosa*, hox—*Hepatico-oxalidosa*, lsp—*Ledo-sphagnosa*, m—*Myrtillosa*, mox—*Myrtillo-oxalidosa*, msp—*Myrtillo-sphagnosa*, ox—*Oxalidosa*, oxn—*Oxalido-nemorosa*, ur—*Urticosa*, v—*Vacciniosa*, vm—*Vaccinio-myrtillosa*.

impossible, even though the climate is suitable (Fig. 1). In such areas, stable communities of this kind are termed edaphic climaxes on immature sand soils or biotic climaxes on hydromorphic soils associated with marshes, swamps or poorly drained uplands (*Petrokas & Manton, 2023*).

According to *Bohn et al. (2000, 2004)* there are seven potential natural forest vegetation types recognised for Lithuania (Table 1). The natural distribution of Norway spruce is mainly driven by climatic conditions. Scots pine forests, partly with deciduous small-leaved tree species and Norway spruce, or Scots pine bog forests should be

**Table 2 Overview of the literature review search criteria outlining each section and the inclusion/exclusion of keywords.**

| Paragraph no. | Search keywords | Inclusion/Exclusion keywords |
| --- | --- | --- |
| Introduction 1 | Closer + To + Nature + Forest | Europe |
| Introduction 2 | Natural + Regeneration ± Genetic + Diversity | Forest ± Tree + Species |
| Introduction 3–4 | Biological + Age ± Plant + Ontogeny | Closer + To + Nature |
| Materials & Methods 1 | Forest + Management | Hemiboreal ± Lithuania |
| Materials & Methods 2 | Climatic + Edaphic + Biotic | Forest + Ecosystem |
| Materials & Methods 3 | Natural + Potential + Vegetation | Hemiboreal ± Climax |
| Materials & Methods 4 | Climax + Vegetation ± Forest + Formation | Hemiboreal ± Lithuania |
| Results 1 | Ontogenetic ± Growth ± Development | Plant ± Tree ± Traits |
| Results 2 | Ontogeny ± Phenology ± Light | Tree ± Regeneration |
| Results 3 | Light ± Photosynthesis ± Respiration | Forest ± Tree ± Vitality |
| Results 4–5 | Gap ± Cohort ± Succession ± Disturbance | Tree ± Stand |
| Results 5–9 | Juvenile ± Immature ± Virginile ± Generative | Forest + Gap ± Light + Intensity ± Shade + Tolerance |
| Discussion 1–2 | Forest + Dynamics ± Ecological + Niche | Forest + Types ± Tree + Species |
| Discussion 4 | Natural + Disturbance ± Silvicultural + System | Natural + Regeneration ± Forest + Dynamics |
| Discussion 5–7 | Natural + Regeneration | Regeneration + Success ± Seed + Production ± Tree + Species |
| Discussion 8 | Closer + To + Nature + Forest | Femelschlag ± System |

considered as edaphic or biotic formations that are in stable equilibrium with their environment. Lithuania's black alder and downy birch carr and swamp forests, classified as swamp and fen forests, are also edaphic or biotic formations, depending on special soil conditions or relief. In these associations one combination of dominants, consisting of stand-forming species with specialized soil requirements or tolerances, seems able to maintain itself permanently. Lithuania's species-rich oak-hornbeam forests, including pine-oak forests, and lime-oak forests are climatic formations, in the development of which climate and vegetation play the principal part.

## Methods of investigation

This review builds on the topic of adaptive relationships in hemiboreal forests as described by *Petrokas & Manton (2023)*. Here, we further develop the subject by exploring the light requirements of early tree growth at different stages of ontogenesis, which have not been previously covered. We reused the same data and methods as applied in *Petrokas & Manton (2023)*. Based on this review, we used Google Scholar, ScienceDirect, and the Web of Science search engines to search for various keywords (Table 2). We did not quantify our research results, in terms of numbers; instead, we analysed the resulting articles until we were able to build a compelling and comprehensive overview summary on the topic of light requirements for the growth and development of individual trees and forests. The criteria for excluding the literature were the quality of the scientific discourse and its relevance to the objectives of the study. Journal metrics or article types (*e.g.*, research article, literature review article, technical report) were not taken into consideration, as the goal of the review was to highlight the importance of understanding the biological age of

trees in the context of closer-to-nature forest management. A Zotero digital bibliographic library was compiled to organise the selected references.

To review the natural regeneration characteristics of hemiboreal forests of northern Europe that reflect the ontogenetic characteristics of stand-forming tree species, we applied a 3-step process.

The first step of the study was to review the ontogenetic characteristics of stand-forming tree species under full and minimal light conditions at the pre-generative and generative stages of ontogenesis (*i.e.*, shape of primary and secondary crowns; branching order of the shoot system; length of annual shoot on the main axis and on lateral branches; *etc.*) based on *Smirnova & Bobrovskii (2001)*, *Evstigneev & Korotkov (2016)*, and *Evstigneev & Korotkova (2019)*. It must be noted that despite the widespread idea of biological age among Russian plant demographers, the definition of calendar age is still dominant in forest science, and few works have been devoted to the problem of the biological age or the ontogenetic (age) spectra of different species of hemiboreal forest trees and their communities. For instance, natural forming stand communities of Norway spruce are generally comprised of mixed species with various ontogenetic dynamics whilst modern management aims towards single species and age stand profiles which lack variation (*Angelstam & Kuuluvainen, 2004*; *Žemaitis, Gil & Borowski, 2019*).

The second step describes the ontogenetic strategies of stand-forming tree species. The four modes of tree establishment and development identified in previous studies (*Petrokas, Baliuckas & Manton, 2020*; *Petrokas & Kavaliauskas, 2022*; *Petrokas & Manton, 2023*) have been adjusted to match the tree light requirements defined by *Evstigneev & Korotkova (2019)*.

Finally, adhering to previous research by *Petrokas & Manton (2023)*, the third step summarizes and discuss the findings of the above steps and proposes a conceptual framework for transitioning towards closer-to-nature management of hemiboreal forest trees.

# RESULTS

## Ontogenetic characteristics of trees

Ontogenesis refers to two distinct but coordinated processes in the life of a plant: differentiation and elongation (*Champagnat, Barnola & Lavarenne, 1986*). Primary growth, that is, the dynamics of metamer emergence, initiated by apical meristems near the tips of shoots (by buds) and roots (by meristematic points), results in the secondary process, the elongation of a plant body. Genetically programmed changes among successive metamers occur as a normal expression of whole-plant ontogeny (*Barthélémy & Caraglio, 2007*). Whole-plant ontogeny encompasses growth, which refers to increases in body size, and development, the allocation of resources to cell differentiation for specialized systems (*Arendt, 1997*). Growth traits of trees include calendar age; height of aboveground parts; stem diameter at breast height and at its base; length of fissuring bark on the trunk; length and width of the crown; branching order of the shoot system; length of annual shoot on the main axis and on lateral branches; and other (*Evstigneev & Korotkov, 2016*). Developmental traits of trees include the presence or absence of juvenile, semi-adult

and adult structures; ability to seed or vegetatively reproduce; ratios of the processes of growth and dying out in shoot and root systems; shape of primary and secondary crowns; and other. When it comes to tree shape and structure, branching is most important. As a result of branching, sibling axes succeed topologically from a parent axis (*Hallé, Oldeman & Tomlinson, 1978*; *Barthélémy, Edelin & Hallé, 1989, 1991*). The main parent axis (stem) is usually taken as the first order of a fixed ontogenetic progression of axis types and the side shoots as the second (*Diggle, 1994*). Within this sequence, spanning from axis 1 to the final axis category and adhering to a defined branching pattern, each branch signifies a specific state of meristematic activity. The collective branch series may be regarded as a comprehensive representation, effectively delivering an overview of the overarching meristematic activity (*Barthélémy, Edelin & Hallé, 1991*).

Tree ontogeny is the genetic trajectories of regenerative processes in trees that repeat in time and space, including both development and reproduction (*DiFrisco, 2019*). Transmissional invariance of the underlying genetic code provides the initial conditions for the next regenerative process (*Balázs, 2014*). The entire chain of regenerative processes, from bud burst to fructification up to seed germination and the formation of the next generation, under changing environmental conditions affects the natural regenerative capacity of tree species (*Gatsuk et al., 1980*; *Petrokas & Kavaliauskas, 2022*). When the species in question is existing in an environment which imposes narrow limits in terms of seasonal changes and climatic factors, phenology determines which competing individual, group of individuals, or species will open its buds late enough to escape late killing frosts or drop seed at the optimal time to increase its chances of germination and survival. It is argued that phenological complementarity can enhance tree survival and therefore forest resistance (*Billing et al., 2022*). Tree size has been found to be a critical factor driving complementarity effects in forests across Europe (*Madrigal-González et al., 2016*). Thus, to explain tree species' phenological patterns across growth phases it is crucial to consider their ontogenetic characteristics (*Ritchie, 1966*; *Segrestin, Navas & Garnier, 2020*). Ontogenetic characteristics of hemiboreal forest trees under full and minimal light conditions at the pre-generative and generative stages of ontogenesis are presented in Table 3. Light is known to be a leading formative factor in forest communities (*Evstigneev & Korotkova, 2019*). The range of light possibilities is the limit of irradiance within which the production process can be carried out (*Evstigneev, 2018*). The lower limit of this range is determined by the minimum light at which tree biomass growth is still possible, and the upper limit is determined by the maximum value of biomass production, which is achieved under conditions of free growth under full light.

In a closed forest community, the reduction of incoming light at any one of several canopy levels below the crowns is one of the most limiting factors for tree survival, growth, and reproduction (*Messier et al., 1999*; *Niinemets, 2010*; *Poorter et al., 2014*). Decreasing the amount of light on a leaf from full sunlight to darkness leads to a change from the gain of organic matter (photosynthesis) to the loss or use of organic matter (dark respiration) (*Parker, 1996*). The unfavourable balance of photosynthesis and dark respiration of the lower and middle branches, as well as of the entire body, defines the low vitality of trees in a closed forest community. In the undergrowth, the umbrella-shaped crown, frequent

**Table 3 Ontogenetic characteristics of hemiboreal forest trees at the pre-generative and generative stages of ontogenesis after natural regeneration is initiated.**

| Ontogenetic stage | Free growth under full light | Low vitality under light minimum |
|---|---|---|
| Juvenile | A distinctive feature of the trees is a single elongated shoot, consisting of several annual increments. The basal part of the shoot lies flat and takes root in some individuals; it is represented by the hypocotyl and a few increments of subsequent years. Individuals of rhizome origin do not have a hypocotyl section, and the length of the shoot developing from adventitious buds reaches high values. Such regenerative shoots are characteristic of most trees, with the exceptions of Qr, Fs, Fe, Ps, and Pa. Leaf blades and needles have a shadow structure. The trees do not extend beyond herbaceous-dwarf shrub layer. | The apical bud of the main (primary) shoot dies, and monopodial growth of the first-order axis stops at the beginning of the $2^{nd}$ or $3^{rd}$ growing season. Nevertheless, the formation of the vertical stem continues typically due to the side bud just above and nearest to the dead one. The central apical shoot is replaced annually by a lateral shoot. Pt and Be individuals under deep shade conditions can live up to 2 years; Qr and Ps–up to 5 years; Pa–up to 6 years; Ug, Cb, and Fe–up to 7 years; Ap–up to 9 years; and Tc–up to 10 years. |
| Immature young | A distinctive feature of the trees is appearance of lateral elongated shoots. Branching and crown formation start; infant crowns with axes of the $2^{nd}$ order are formed. The anatomy of leaves still retains the shadow structure. | Substantial shading defines replacement of the central apical shoot by a lateral shoot in the shoot system. The $3^{rd}$ branching order is formed. Pt and Be individuals under light starvation can live up to 7 years; Ps and Pa–up to 10 years; Ug–up to 12 years; Qr, Ap, Fe, and Tc–up to 16–18 years; and Cb–up to 29 years. |
| Immature | The current height increment is several times greater than the lateral increment; the upper part of the crown, which enters the shrub layer, becomes narrow and elongated, the lower part is usually broad because it is an artefact of the umbrella-shaped crown that was formed in the herbaceous-dwarf shrub layer. The $3^{rd}$ branching order is formed. | The growth of the shoots is reduced to a minimum, and the side branches are catching up to the height of the leader axis. The branching order increases in the crown as well as in individual shoots, and some branches die. A system of 'stumps' with adventitious roots is formed at the base of the tree. Pt and Be individuals under the canopy can spend 11–12 years; Ug and Qr–14 years; Cb–19 years; Tc–21 years; Ps–22 years; Fe–23 years; Pa–25 years; Ap–32 years. |
| Virginile young | Upper lateral shoots deviate from the trunk at an acute angle for most tree species. The annual growth rates of the upper lateral shoots are 2–3 times less than that of the leader axis. Intensively branching trees (*e.g.*, Cb and Qr) have the $5^{th}$ branching order, less intensively branching trees (*e.g.*, Ap and Fe)–the $4^{th}$. Be individuals can produce seed in 6 years; Pt–in 8 years; Cb, Ap and Ps–in 11 years; Fe, Pa, and Qr–in 12–13 years; and Ug and Tc–in 16 years. | The umbrella-shaped crown, frequent replacement of the central apical shoot by lateral shoots, cleaning of the trunk from the lower branches and minimal annual increments are signs that the tree has reduced its growth processes. Pt and Ps trees do not have clearly expressed umbrella-shaped crowns. Be individuals under the canopy can spend 12 years; Pt–16 years; Ps–18 years; Cb–22 years; Qr and Tc–24 years; Ap–37 years; Fe, Ug, and Pa–48 years; and Fs–70 years. |
| Virginile | Virginile trees occupy a place in the tree layer. The height increment is greater than at any other ontogenetic stage. Intensively branching trees (*e.g.*, Cb and Qr) have the $6^{th}$ branching order, less intensively branching trees (*e.g.*, Ap and Fe)–the $5^{th}$. The age of individuals is up to 25 years. | The trunk is clean from the lower branches at a considerable height; the multilayer crown is converted to a single-layer type, sometimes to umbrella-shaped. The branching order of shoot system increases. The age of Pt individuals is up to 18 years; Be and Ug–up to 25 years; Qr and Cb–up to 29 years; Tc, Ap, and Fe–up to 40 years; Ps–up to 60 years; and Pa–up to 75 years. |
| Generative young | The cortex begins to form in the lower part of the trunk. Intensively branching trees have the $7^{th}$ branching order, less intensively branching trees–the $6^{th}$. The fruits of flowering trees and seed cones of gymnosperms are usually in the upper half of the crown. | The stag-headed trees; minimal growth of the main axis and large trunk diameters. |
| Generative mature | A domed crown is a distinctive feature of most trees. The bark with deep cracks in different tree species expands to a height equal to up to two-thirds of the trunk height. Intensively branching trees have the $8^{th}$ branching order, less intensively branching trees–the $7^{th}$. Seed production is the highest. | Small crown diameters and dead branches in the lower part of the crowns. |

| Table 3 (continued) | | |
| --- | --- | --- |
| Ontogenetic stage | Free growth under full light | Low vitality under light minimum |
| Generative old | The growth in height of the trees stops, and the upper part of the trunk dries out and dies; secondary crown formation starts (except Ps). The trees typically have a flat-topped crown with dead skeletal branches. The bark with deep cracks all over the trunk. Intensively branching trees have the 9th branching order, less intensively branching trees–the 8th. | Piped rot can occur in trees. |

**Note:**
Ap—*Acer platanoides* L., Be—*Betula pendula* Roth, Cb—*Carpinus betulus* L., Fs—*Fagus sylvatica* L., Fe—*Fraxinus excelsior* L., Pa—*Picea abies* L. Karst, Ps—*Pinus sylvestris* L., Pt—*Populus tremula* L., Qr—*Quercus robur* L., Tc—*Tilia cordata* Mill., Ug—*Ulmus glabra* Huds.

replacement of the central apical shoot by lateral shoots, cleaning of the trunk from the lower branches and minimal annual increments are signs that the plant has reduced its growth processes and expects improved light conditions at a low level of vitality (*Evstigneev & Korotkov, 2016*). With a constant minimum light intensity, tree development can linger for many years in each of the ontogenetic stages. A multilayer crown, which extends vertically, and large annual growth increments with dense leaves are evidence of active growth processes (*Evstigneev & Korotkova, 2019*). Thus, the light balance within the multiple canopies can be considered a necessary condition for the transition of individuals to subsequent ontogenetic stages. This suggests that forest management operations such as forest thinning should try to maintain natural light balance within multiple canopies.

## Ontogenetic strategies of trees

Closer-to-nature forest management aims towards promoting assisted natural regeneration of trees by establishing favourable environmental and ecological conditions for their successful growth, and development (*European Commission, 2023*). A better understanding of the relation between overstory gaps, understory vegetation dynamics, and the ontogeny of stand-forming tree species can help develop natural regeneration-based management methods for different forest types.

The classical understanding of "gap" in forest science is within the gap dynamics paradigm: a gap in the forest canopy resulting from the death of a tree or group of trees or stand replacement in response to disturbance. The hemiboreal forest zone contains three broad forest disturbance regimes of different sizes, i) small tree-scale gap associated to a single tree to a small group of trees, ii) multi-cohort dynamics with partial stand-scale disturbances and iii) successional stand development in response to landscape-scale disturbance events resulting in even-aged stands (*Angelstam & Kuuluvainen, 2004*; *Kuuluvainen, 2016*). The disturbance regime is simply a description of the types of disturbance characteristic of a given forest landscape; the scale and agents of these disturbance types are the primary considerations in predicting the kind, quantity, and spatial pattern of biological legacy (*Frelich, 2002*; *Franklin, Mitchell & Palik, 2007*). However, the distinction of each disturbance regime can be fuzzy depending on the tempol and spatial scale as well as the intensity of the disturbance and species composition. For instance, the succession regime with large disturbances may also develop multi-cohort

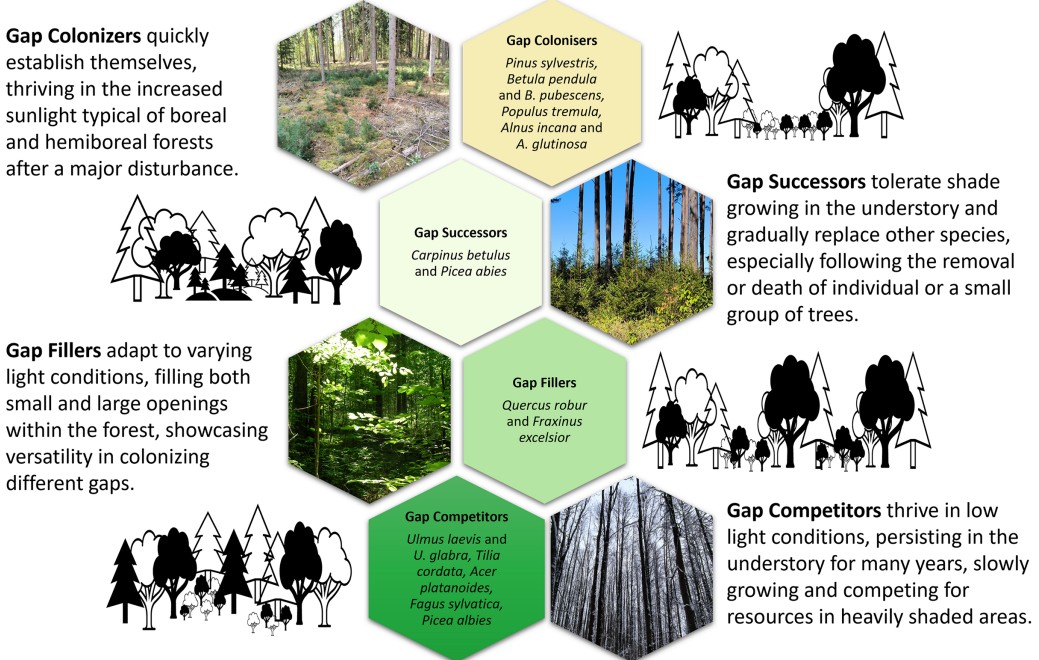

**Gap Colonizers** quickly establish themselves, thriving in the increased sunlight typical of boreal and hemiboreal forests after a major disturbance.

**Gap Colonisers**
*Pinus sylvestris, Betula pendula* and *B. pubescens, Populus tremula, Alnus incana* and *A. glutinosa*

**Gap Successors** tolerate shade growing in the understory and gradually replace other species, especially following the removal or death of individual or a small group of trees.

**Gap Successors**
*Carpinus betulus* and *Picea abies*

**Gap Fillers** adapt to varying light conditions, filling both small and large openings within the forest, showcasing versatility in colonizing different gaps.

**Gap Fillers**
*Quercus robur* and *Fraxinus excelsior*

**Gap Competitors**
*Ulmus laevis* and *U. glabra, Tilia cordata, Acer platanoides, Fagus sylvatica, Picea albies*

**Gap Competitors** thrive in low light conditions, persisting in the understory for many years, slowly growing and competing for resources in heavily shaded areas.

**Figure 2 An conceptual overview of ontogenetic strategies of tree regeneration for European hemiboreal forests.** Photos and design by D. Kavaliauskas, M. Manton and R. Petrokas. Graphic design components are royalty-free from the MS Office stock.

dynamics and or small-scale gaps at any given time (*Kuuluvainen, Bergeron & Coates, 2015*). Within each of these broad forest disturbances regimes, the response of hemiboreal forest trees differs in regeneration and growth modes (*Vodde et al., 2010*). The analysis of characteristics of trees at the pre-generative and generative stages of ontogenesis under full and minimal light conditions allows these differences to be named as four ontogenetic strategies of stand-forming tree species.

First, **gap colonisers** (Fig. 2) have low shade tolerance combined with rapid development, high growth intensity and physiological processes (photosynthesis and respiration), with large average annual biomass increases and a short lifespan when growing in the undergrowth due to a lack of light (*Evstigneev & Korotkova, 2019*). This set of characteristics allows these trees to colonise large treefall gaps (frequently with exposed mineral soils) as well as sparse forests and grow in them as dominants (*Yamamoto, 1996*; *Petrokas, Ibanga & Manton, 2022*). Eurasian aspen, silver birch and Scots pine are characterised by a highest light demand and low shade tolerance; they develop to the juvenile stage at light intensity of 2.7%, 3.1% and 6.0% of full light in the open but reach immature stage at 6.6%, 9.6% and 13.9% and the virginile stage only at 17.9%, 23.0% and 34.3%, respectively (*Evstigneev & Korotkova, 2019*). Grey alder is a more light-demanding species compared to black alder, which is replaced by other species as soon as the canopy closes (*McVean, 1953*).

Second, **gap successors** appear in forest sites with high light conditions, and saplings can survive under a closed canopy (*Evstigneev, 2018*; *Petrokas, Ibanga & Manton, 2022*).

Norway spruce and European hornbeam are among the most shade-tolerant species; they develop to the juvenile stage at light intensity of 1.1% and 0.7% of full light in the open but reach the immature stage at 1.2% and 1.5% and the virginile stage at 1.4% and 1.8%, respectively (*Evstigneev & Korotkova, 2019*). The European hornbeam grows primarily below the canopy of other broadleaves, such as light-loving English oak (*Kuehne et al., 2014*; *Evstigneev, 2018*). Norway spruce is the most common admixture in stands of other dominants (*Kuliešis et al., 2021*).

Third, **gap fillers** survive in newly created light gaps (*Petrokas, Ibanga & Manton, 2022*). At the juvenile stage they are shade-tolerant, but at subsequent stages of ontogenesis their need for light increases (*Evstigneev, 2018*). European ash and English oak develop to the juvenile stage at light intensity of 0.4% and 1.2% of full light in the open but reaches immature stage at 0.9% and 4.5% and the virginile stage at 4.2% and 10.4%, respectively (*Evstigneev & Korotkova, 2019*). The ability of English oak to change its life state is the most important mechanism of adaptation to the constantly changing light conditions of the forest environment (*Evstigneev & Korotkova, 2024*).

Fourth, **gap competitors** are adapted to habitats beneath a dark forest canopy formed by spruce and broadleaved trees with crowns that cast deep shadow (*Evstigneev & Korotkova, 2019*). They are represented by Norway maple, small-leaved lime and wych elm and are characterised by the greatest shade tolerance at all ontogenetic stages; they develop to the juvenile stage at light intensity of 0.3%, 0.6% and 0.5% of full light in the open but reach immature stage at 0.5%, 0.8% and 0.7% and the virginile stage at 0.8%, 1.0% and 1.1%, respectively (*Evstigneev & Korotkova, 2019*). European beech is the most shade-tolerant broadleaved tree and the strongest competitor in its range (*Walter, 2012*).

In summary, gap colonisers proposed by us are closest to the ruderal (reactive) type of strategy and gap competitors to the tolerant type, according to *Smirnova (1994)*. Whereas gap successors and gap fillers are represented by tree species that are most often attributed to the competitive strategy. Thus, the four strategies can be arranged from large gaps to medium gaps, small gaps, and understory growth, with the latter two sometimes also responding by release from suppression makes sense.

## DISCUSSION

### Managing forest tree regeneration and growth

The adaptive capacity and resilience of forest communities is determined by plant regeneration strategies to deal with unique environmental conditions related to competition, abiotic limitation to growth (stress) and periodic destruction of biomass (disturbance) (*van Schaik, Terborgh & Wright, 1993*; *Yang & Rudolf, 2010*; *Dayrell et al., 2018*). Our conceptual framework for transitioning toward closer-to-nature management of hemiboreal forest trees posits that there are complex adaptive dynamic relationships in hemiboreal tree communities. It promotes natural regeneration that matches the biological age dynamics of stand-forming tree species, and protects the retention of the natural ecological patterns and processes of the northern European hemiboreal forests. It follows the Lithuanian classification of forest types, four types of stand-forming tree species, three

**Table 4** A conceptual framework for transitioning towards closer-to-nature management of hemiboreal forest trees.

| Forest type series** | Regeneration strategies of trees* | | | | Forest dynamics | Disturbances | | Silvicultural system |
|---|---|---|---|---|---|---|---|---|
| | Gap colonisers | Gap successors | Gap fillers | Gap competitors | | Abiotic | Biotic | |
| oxn | Pt Be Bu Ai Ag | Pa | Qr Fe | Tc Ug Ap | Gap-phase | Windthrow | Diseases, insects, clear-cutting | Single tree selection |
| ox, mox | Ps Pt Be | Pa | Qr | – | Even-aged | Fire, windthrow | | Irregular shelterwood |
| vm, m | Ps Be Pt | Pa | – | – | Even-aged | Fire | Soil compaction | Irregular shelterwood |
| v | Ps Be | – | – | – | Multi-cohort | Fire | Soil compaction | Group selection |
| cl | Ps | – | – | – | Multi-cohort | Fire | Soil compaction | Group selection |
| msp | Ps Bu | – | – | – | Even-aged | Fire | | Irregular shelterwood |
| csp | Ps Bu | – | – | – | Gap-phase | Windthrow | Clear-cutting | Group selection |
| lsp | Ps | – | – | – | Even-aged | Fire | Physiological draught | Irregular shelterwood |
| hox | Pt Be Ai Ps | Pa Cb | Qr | Fs Tc Ug Ul Ap | Gap-phase | Windthrow | Diseases, insects, clear-cutting | Single tree selection |
| aeg, cmh | Pt Be Bu Ag Ai | – | Qr Fe | Tc Ug Ul Ap | Gap-phase | Windthrow | Diseases, insects, clear-cutting | Single tree selection |
| fil, ur | Ag Bu Be Ai | Pa | Fe | – | Gap-phase | Windthrow | Clear-cutting | Group selection |
| cir | Ag Bu | Pa | – | – | Gap-phase | Windthrow | Clear-cutting | Group selection |
| c | Bu Ag | Pa | – | – | Gap-phase | Windthrow | Clear-cutting | Group selection |
| cal | Bu Ag Be | Pa | Qr | – | Even-aged | Fire, windthrow | | Irregular shelterwood |

**Note:**
*Ag—*Alnus glutinosa* L. Gaertn., Ai—*Alnus incana* L. Moench, Ap—*Acer platanoides* L., Be—*Betula pendula* Roth, Bu—*Betula pubescent* Ehrh., Cb—*Carpinus betulus* L., Fs—*Fagus sylvatica* L., Fe—*Fraxinus excelsior* L., Pa—*Picea abies* L. Karst, Ps—*Pinus sylvestris* L., Pt—*Populus tremula* L., Qr—*Quercus robur* L., Tc—*Tilia cordata* Mill., Ug—*Ulmus glabra* Huds., Ul—*Ulmus laevis* Pall. **Ground layer codes of the main types of plant communities (*Karazija, 1988*): aeg—*Aegopodiosa*, c—*Caricosa*, cal—*Calamagrostidosa*, cir—*Carico-iridosa*, cl—*Cladoniosa*, cmh—*Carico-mixtoherbosa*, csp—*Carico-sphagnosa*, fil—*Filipendulo-mixtoherbosa*, hox—*Hepatico-oxalidosa*, lsp—*Ledo-sphagnosa*, m—*Myrtillosa*, mox—*Myrtillo-oxalidosa*, msp—*Myrtillo-sphagnosa*, ox—*Oxalidosa*, oxn—*Oxalido-nemorosa*, ur—*Urticosa*, v—*Vacciniosa*, vm—*Vaccinio-myrtillosa*.

modes of forest dynamics, and four types of natural disturbance-based silvicultural systems (*Hawley & Smith, 1954*; *Karazija, 1988*; *Navasaitis et al., 2003*; *Angelstam & Kuuluvainen, 2004*; *Barbati, Corona & Marchetti, 2006*; *Aakala et al., 2011*; *Myking et al., 2011*; *Gabrilavičius, Petrokas & Danusevičius, 2013*; *Kuuluvainen, Bergeron & Coates, 2015*; *Kuuluvainen, 2016*; *Plesa et al., 2018*; *Stobbe & Gumnior, 2021*; *Armolaitis et al., 2022*; *Petrokas & Manton, 2023*; *Gonçalves & Fonseca, 2023*) (Table 4).

Ecological niche is the subset of environmental conditions that affect a specific population of trees (*Chamary, 2023*). Mixed tree communities with high functional diversity, where broadleaf and coniferous trees coexist, contain a broad range of functional niches, which can lead to more efficient resource use and adaptation to environmental changes (*Billing et al., 2022*). A summary of our analysis of the ontogenetic characteristics of hemiboreal trees shows that stand-forming species occupy one to several niche positions relative to forest dynamics modes. For instance, the niche position of European hornbeam is restricted to the gap-phase processes that typically operate throughout stand

development caused by the death of individual trees or small groups of trees in English oak-hornbeam forests (*e.g.*, *Hepatico-oxalido-Quercetum/Carpinetum* forest types). In contrast, the niche position of Scots pine can be characterised as even-aged dynamics in mixed Norway spruce forests (*e.g.*, *Oxalido/Myrtillo-oxalido-Piceetum/Pinetum* forest types), multi-cohort dynamics in Scots pine forests (*e.g.*, *Vaccinio/Cladonio-Pinetum* forest types), and gap-phase to multi-cohort dynamics related in Scots pine bog forests (*e.g.*, *Carico-sphagno-Pinetum* forest type).

Moreover, English oak and European ash are difficult to regenerate in light restricted forests, this has been controversial issue for more than a century (*Götmark et al., 2005*). The extreme viewpoint is the assumption of a "non-forest" biology of these species and a close connection between their regeneration and grazing by wild ungulates (Frans Vera theory; *Angelstam et al. (2017)*). *Smirnova, Bobrovsky & Khanina (2001)* proposed to identify a special forest edge group of tree species along with early-successional and late-successional species. It includes species whose successful regeneration is associated with meadows and edges following grazing and browsing or tree seed which are animal-dispersed. These includes species such as wild apple, wild pear, and English oak. Currently, the main place of natural regeneration of English oak is abandoned meadows and secondary light forests. The situation with understanding European ash regeneration is even more complicated and it is impossible to summarize it briefly. However, recent studies show (*e.g.*, *Shashkov et al., 2022*) a different result: wind-dispersed ash trees successfully spread to abandoned arable land and pastures (*Smirnova, Bobrovsky & Khanina, 2018*). This brief overview shows that each species' uniqueness is worthy of further research.

None the less most disturbances are either driven by/or are indirectly linked to environmental conditions related to abiotic limitation to growth and periodic destruction of biomass (*Boisvenue & Running, 2013*; *Dayrell et al., 2018*). Incorporating an understanding of natural disturbance and forest dynamics more fully into silvicultural practice is the basis for an ecological forestry approach (*Franklin, Mitchell & Palik, 2007*). The goal of ecological forest management is to obtain the maximum number of planned sizes and quality assortments while strictly adhering to all environmental and biodiversity requirements. In this model, between 5% and 30% of all grown wood is usually lost in the form of dead trees, but what is the most important is that silviculture under ecological forestry can mimic natural disturbance severity and return intervals and provide a complete range of habitats (*McCoy & Bell, 1991*; *Molefe, 2019*; *Kuliešis et al., 2021*; *Himes et al., 2022*). Thus, due to disturbance regimes and stand development processes, the choice of silvicultural system is critical to the success of natural regeneration. Unlike traditional shelterwood and seed tree applications, most natural disturbances leave a living legacy, such as resilient mature overstory trees of varying sizes, as well as intact layers of understory (including seedling banks). Therefore, silvicultural interventions should foster adaptability and connectivity among populations to facilitate gene flow and species migration (*Himes et al., 2022*). In general, multi-aged, gap-based silvicultural methods, including group selection, single tree selection, and expanding-gap irregular shelterwood, aim to emulate natural disturbance regimes and the dynamics of complex forest structure

under active management (*Seymour, White & deMaynadier, 2002*; *Himes et al., 2022*). The group shelterwood benefits more shade-tolerant and shade-intermediate species while it is inappropriate for shade-intolerant species or species sensitive to frost damage (*Hawley & Smith, 1954*). Greater regeneration success of shade-intolerant species is achieved in group selection (*Gonçalves & Fonseca, 2023*). As a rule, in hemiboreal landscapes, indirect evidence of stand-replacing disturbances or silvicultural interventions such as clear-cutting are young even-aged forest stands of shade-intolerant species, such as birches, Eurasian aspen, and Scots pine (*Shorohova et al., 2009*). Moreover, in modern hemiboreal forests it is very rare to find forests with close to climax dynamics with a variety of natural gap dynamics. This leads to the need for assisted natural regeneration of trees to emulate the natural dynamics of forest structure.

## Assisted natural regeneration of trees

Assisted natural regeneration lays the groundwork necessary to consider the life-cycle features of trees that affect the complex adaptive dynamic relationships in hemiboreal tree communities indirectly *via* their effects on survival, growth, and reproduction (*Petrokas, Ibanga & Manton, 2022*). Natural regeneration of trees is a function of several sequential ecological processes, pollination and fertilization, seed development and maturation, seed predation, dispersal and germination, seedling establishment, vegetative growth, natural selection, and biological maturity. Successful reproduction and dispersal are the first step in landscape-level forest dynamics and, eventually, in the way forest regenerates after disturbance (*Boisvenue & Running, 2013*). Processes such as annual seed production, seed dispersal, and environmental conditions for germination, early survival, and early growth of regenerating trees can cause large differences in regeneration success (*De Lombaerde, 2020*). Seed dispersal and early seedling recruitment establish the foundation for plant regeneration and can significantly impact the demography and evolution of plant populations (*Clark et al., 1999*; *Nathan & Muller-Landau, 2000*; *Hampe, El Masri & Petit, 2010*; *Bontemps, Klein & Oddou-Muratorio, 2013*). Overall, regeneration success can be quite variable due to edaphic and climatic conditions, seed losses, and/or seedling mortality (*Gonçalves & Fonseca, 2023*). *Den Ouden et al. (2010)* identified four critical phases that must be passed to achieve successful natural regeneration: (1) seed production (*e.g.*, synchrony in flowering phenology, successful pollination and seed formation), (2) seed fall or presence in the seed bank at the time regeneration is initiated, (3) germination, emergence, and establishment, and (4) survival and growth. In addition, the variability in seed production between years, known as regular or irregular masting, mast seeding, seed predators, seed dispersal models and distances, have a significant impact on natural regeneration (*McVean, 1955*; *Matlack, 1987*; *Venable & Brown, 1988*; *Kelly & Sork, 2002*; *Heuertz et al., 2003*; *Kramer, Bruinderink & Prins, 2006*; *Kutter & Gratzer, 2006*; *Barbour & Brinkman, 2008*; *Harrington et al., 2008*; *Navasaitis, 2008*; *Övergaard, 2010*; *Claessens et al., 2010*; *Pigott, 2012*; *Bontemps, Klein & Oddou-Muratorio, 2013*; *Venturas, Nanos & Gil, 2014*; *Evstigneev & Korotkov, 2016*; *Beck et al., 2016*; *Gerzabek, Oddou-Muratorio & Hampe, 2020*; *Liu & Evans, 2021*) (Table 5). For instance, species such as black alder (*McVean, 1955*), European ash (*Eisen et al., 2024*), and European white elm

**Table 5 Some seed-bearing and seed dispersal characteristics of hemiboreal forest trees.**

| Tree ontogenetic strategies | Tree species | Seed-bearing age (years) | | Seed harvest interval | Seed dispersal models |
|---|---|---|---|---|---|
| | | Forest | Open | | |
| Gap colonisers | *Alnus glutinosa* | 30[1] | 6[2] | Every 3–4 years[1] | Wind and water |
| | *Alnus incana* | 25[2] | 5[2] | Every 1–4 years[2] | Wind and water |
| | *Betula pendula* | 25[3] | 6[3] | Annually (more intensively every 2–3 years)[4,5] | Wind |
| | *Betula pubescens* | 25[5] | 10[5] | Annually (more intensively every 2–3 years)[5] | Wind and water |
| | *Pinus sylvestris* | 40[3,4] | 11[3,4] | Every 3–5 years[4] | Wind |
| | *Populus tremula* | 25[3,4] | 8[3] | Annually[4] | Wind |
| Gap successors | *Carpinus betulus* | 30[3] | 11[4,3] | Almost every year[4] | Wind |
| | *Picea abies* | 50[3] | 13[3] | Every 3–5 (6) years[4] | Wind |
| Gap fillers | *Fraxinus excelsior* | 45[3] | 12[3] | Annually (more intensively every 2–3 years)[4,5] | Wind and water |
| | *Quercus robur* | 60[3,4] | 13[3] | Every 3–4 years[4] | Animals and birds |
| Gap competitors | *Acer platanoides* | 40[3] | 11[3] | Annually[4] | Wind |
| | *Fagus sylvatica* | 70[6] | 40[6] | Every 3–5 years[6] | Wind???, animals and birds |
| | *Tilia cordata* | 40[3] | 16[3] | Annually[4] | Wind |
| | *Ulmus glabra* | 30[3] | 15[3,4] | Almost every year[4] | Wind |
| | *Ulmus laevis* | 30[7] | 10[4] | Annually (more intensively every 2–3 years)[1] | Wind and water |

**Note:**
*References: [1]*Claessens et al. (2010)*, [2]*Harrington et al. (2008)*, [3]*Evstigneev & Korotkov (2016)*, [4]*Navasaitis (2008)*, [5]*Beck et al. (2016)*, [6]*Övergaard (2010)*, [7]*Barbour & Brinkman (2008)*.

(*e.g.*, *Venturas, Nanos & Gil, 2014*) often rely on water for seed dispersal, which can be a limiting factor for dispersal and germination. Furthermore, species with heavy seeds like English oak and European beech often depend on animal-mediated seed dispersal and seed predator's satiation (*Kelly & Sork, 2002*; *Bontemps, Klein & Oddou-Muratorio, 2013*). Thus, mixture of tree species with different seed dispersal models enhances the uniformity of seed distribution across patches, a process that is indirectly influenced by changes in germination and seed size. Typically, seeds are predominantly produced in patches with above-average densities. Consequently, uniform dispersal leads to a net movement of seeds to patches with lower-than-average density (*Venable & Brown, 1988*).

Furthermore, the age of seed bearing, which is varied between species and trees in different environments (open landscapes and forests (Table 5)), cannot justify harvesting at a younger age, since the new generation of forest trees is effectively established based on the reproductive cycles of the species and the waiting years for the mast seeding (*Petrokas & Kavaliauskas, 2022*). Therefore, mast seeding should be used throughout the life of the forest to allow natural regeneration and/or silvicultural practices that promote the utilisation of the seed source (*e.g.*, *Övergaard, 2010*; *Gärtner, Lieffers & Macdonald, 2011*; *Pearse et al., 2021*). For example, in mast years, the presence of seed-caching and berry-eating species is critical to the distribution of animal-dispersed oak trees, which depend on their seeds being carried away from the parent tree and buried by jays, mice, and squirrels (*Moran, 2019*).

The impoverished composition of many modern forests is the result of a long land-use history that has created gaps in the local ranges of tree species (*e.g.*, *Johann et al., 2004*; *Myking et al., 2016*; *Jansen, Konrad & Geburek, 2019*). In many cases, successful spread of species is hindered by short dispersal distances compared to the large size of anthropogenic disturbances. In any case, due to biological reasons and various forest management approaches, the dynamics of natural regeneration and growth of forest trees is quite difficult to modify. After all, the seeds of many species of trees have special adaptations that allow them to remain dormant for years, waiting for optimal conditions to germinate and effectively establish a new generation (*Venable & Brown, 1988*; *Klupczyńska & Pawłowski, 2021*).

### Incorporating closer-to-nature forest management

During this review we have also found that promoting closer-to-nature forest management through the emulation of natural disturbance-based patterns and processes is not an entirely new concept or approach. In 1880, Gayer developed the "Femelschlag" forest management approach (*Silvy-Leligois, 1953*; *Raymond et al., 2009*), a traditional reproduction technique used to produce multi-species forest stands by promoting shade intolerant trees, such as English oak, Scots pine, *etc* (*Puettmann et al., 2009*). It aims at harnessing accelerated natural regeneration by emulating natural disturbance-based patterns (gap size) and encouraging tree species diversity in multiple-age classes, thereby enhancing ecosystem complexity and resilience (*Mohr & Schori, 1999*). Targeted removal of the tree canopy allows more sunlight to reach the forest floor stimulating seeds stored on the ground to regeneration in the gaps, creating mixture of old and young forest conditions. Thus, moving towards closer-to-nature forest management involve wholistic analysis of past management systems combined with natural ecological patterns and processes of forest trees, vegetation communities (such as in this study), as well as novel methods of balanced and sustainable forest management.

## CONCLUSIONS

This study shows the importance of understanding tree ontogeny in the context of closer-to-nature forest management in Europe, and its potential towards formulating sustainable practices that emulate the natural dynamics of forest structure that assist gene flow and species migration and help adapt and mitigate climate change. By recognizing and respecting the growth and developmental status of individual trees and forests, forest managers can implement strategies that align with the inherent dynamics of the ecosystem. This approach prioritizes ecological integrity and forest resilience, fostering a more harmonious coexistence between human activities and the natural environment. Using the hemiboreal forests of northern Europe, we have demonstrated the specific developmental characteristics of individual trees and forests so that scientists and foresters can tailor management strategies to enhance overall forest health, adaptive capacity, resilience, and productivity. This knowledge can be used to determine tree species composition and thus aid in developing closer-to-nature forest management practices, enabling the conservation of habitats necessary for the dynamic functioning of the ecological processes essential to

the regeneration of the natural resources, optimal timber harvesting, and the mitigation of environmental impacts. In the pursuit of effective natural resource stewardship, an in-depth understanding of the ontogenetic stages of forest trees becomes indispensable towards fostering the development of resilient and thriving ecosystems. The priority now is to regenerate the forest naturally, with supplementary planting, where necessary. Letting nature lead is one of the key principles of emulating nature, with tree planting coming last. This review shows that understanding the interplay of climatic, edaphic, and biotic factors is a crucial first step towards developing and implementing ecological forest management strategies that mimic natural disturbance severity and return intervals and provide a complete range of habitats.

## ACKNOWLEDGEMENTS

This review presents research findings that have been obtained through the long-term research program "Sustainable forestry and global changes", implemented by the Lithuanian Research Centre for Agriculture and Forestry.

### Funding

The research contribution of Michael Manton was carried out within the framework and funding of the European Union's Horizon Europe programme Eco2adapt, grant agreement No. 101059498. The funders had no role in study design, data collection and analysis, decision to publish, or preparation of the manuscript.

### Grant Disclosures

The following grant information was disclosed by the authors:
European Union's Horizon Europe programme Eco2adapt: 101059498.

### Competing Interests

The authors declare that they have no competing interests.

### Author Contributions

- Raimundas Petrokas conceived and designed the experiments, performed the experiments, analyzed the data, prepared figures and/or tables, authored or reviewed drafts of the article, and approved the final draft.
- Michael Manton conceived and designed the experiments, performed the experiments, analyzed the data, prepared figures and/or tables, authored or reviewed drafts of the article, and approved the final draft.
- Darius Kavaliauskas performed the experiments, analyzed the data, prepared figures and/or tables, and approved the final draft.

### Data Availability

  This is a literature review.

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
