# Peer review of "Tree regeneration and ontogenetic strategies of northern European hemiboreal forests: transitioning towards closer-to-nature forest management"

_PeerJ, doi:10.7717/peerj.17644_

## Round 0.1 · original submission · Major Revisions

Dear Dr. Petrokas,

I have now received the reviews of your paper. As you can see, three of the reviewers suggested minor changes, whereas one reviewer requested more substantial changes.

My main concern with your paper is the great similarity with some of your other published works. Turnitin gave a very high score of 33%. I urge you to rephrase parts of the text showing similarity with already published works in the revised version.

Sincerely,
Shaw Badenhorst

·

Basic reporting

The article by Raimundas Petrokas, Michael Manton and Darius Kavaliauskas is devoted to the review of ideas on discrete description of tree ontogeny and tree regeneration strategies for the development of closer-to-nature forest management strategies. The contributions are at the interface between population ontogenetic studies, forest ecology and forestry. They form bridges between forest research at different scales and forestry practice. This review is of broad interest and within the scope of the journal.
Attempts to integrate insights from ontogeny and tree biology into forest practice to develop ecosystem-based forest management have been made repeatedly since the late 1980s. However, this topic has not been exhausted, primarily because of the intensive development of ecology in recent decades. New ecological concepts allow a new approach to this task.
The introduction presents in detail and adequately the topic of the review and its relevance for a wide range of specialists related to both forest theory and practice.

Experimental design

The research methodology is generally consistent with the broad coverage of the chosen topic. The presentation of results and discussion is partly consistent and logical, but requires improvement. The complexity of the article lies in the comprehensiveness of the conception concerning many issues. The links between the different parts of the review (descriptions of ontogeny features, Tree Regeneration Strategies and the discussion) could be significantly improved.
The results section could be improved by more fully citing sources on the main topics: discrete model descriptions of tree ontogeny, population and regeneration strategies of tree species. The discussion needs a fuller link to the results and a clearer discussion of the tree biology facts cited. It is also desirable to provide examples of how ontogenetic development and regeneration patterns are realized under different conditions: in different forest types, under different forest dynamics models, after different disturbance options. This is necessary for a clear understanding of the links between the ideas outlined and closer-to-nature forest management.

Validity of the findings

The conclusion and abstract are generally true to the content of the review, identifying opportunities for future directions for forest management in relation to more fully engaging knowledge of tree species biology. However, the conclusion is now, in part, largely a promise that requires a more extended justification in the discussion.

Additional comments

The following are some comments and suggestions for the article improvement.
1. Lines 199-213: it is an excursion into metameric structure, meristem operation, and the order of axis formation. It is not clear what relevance many of the issues discussed have to the task of the paper. These concepts are not used further. It is possible to shorten this part.
2. Line 214-264: it is a review of ideas of discrete description of ontogenesis, age stages and peculiarities of tree species ecology. The material is presented in a consistent and understandable manner. However, this review is built mainly on a few articles by Evstigneev et al, which are themselves reviews. As a result, the reader gets a distorted view of the authors of the concepts and the main literature sources. I think it is highly desirable to provide references to the following major works.
The described approach with respect to trees was developed and widely used thanks to Prof. Olga Smirnova and her students (who also include Oleg Evstigneev and Vladimir Korotkov). In English, the concept of discrete description of ontogenesis was firstly introduced in the article by Gatsuk et al., 1980 (Age states of plants of various growth forms: a review), where, among other things, the scheme of ontogenesis of Fraxinus excelsior is given by Ludmila Zaugolnova. A generalizing article in English on the ontogeny of trees is Smirnova et al., 1999 (Ontogeny of a tree). Brief descriptions and examples of use are given in the articles Smirnova et al., 2000 (Population mosaic cycles in forest ecosystems, IAVS Proc.); Smirnova, Bobrovsky, 2001 (Tree ontogeny and its reflection in the structure and dynamics of plant and soil covers, Rus.J.Ecol.), and in the book "European Russian Forests. Their Current State and Features of Their History" (2017) (ed. by O.Smirnova, M. Bobrovsky, and L.Khanina).
Among publications in Russian, a very important one is the book "Eastern European Broadleaved Forests" (editor Smirnova, 1994; https://www.researchgate.net/publication/323692492_Vostocnoevropejskie_sirokolistvennye_lesa_Pod_red_OV_Smirnovoj_M_Nauka_1994_364_s), which for the first time provided the most complete descriptions of the ontogeny of the main tree species of Eastern Europe and data on their biology (including relationship to light, population strategies, and much more).
The theoretical discussion of the population approach to the analysis of different ecosystems was first outlined most fully by Olga Smirnova, 1998 (Population organization of the biocoenotic cover of forest landscapes, in Russian).
3. An important issue not addressed by the authors of the peer-reviewed manuscript is the use of the concept of ontogenetic stages to analyze the ontogenetic (age) spectra of different species. This application is the main "bridge" from the level of the individual to the level of the population and to the level of the community (ecosystem). Comparative analysis of age spectra of tree species is widely used to assess the successional stage of a plant community. This approach has been implemented in a large number of studies (mostly in Russian). In English, the above-mentioned book "European Russian Forests" (2017) presents results of applying this approach to analyze boreal, hemiboreal, and nemoral forests. It seems to be a valuable example of the importance of considering the ontogenesis of different tree species to assess the successional stage of communities and to understand the role of the population approach in the study of forest dynamics. The distribution of individuals of different ontogenetic stages across forest layers in different forest stands is the basis for the classification of regeneration strategies of tree species.
4. Lines 266-309: The authors build a classification of renewal strategies based on a previously proposed classification (Petrokas & Manton, 2023), but it raises some questions. Probably the change in the composition of some groups from the previous publication also requires clarification.
The authors need to formulate more clearly the criteria for grouping; compare them with other studies. It is not clear from the proposed text which treats are the main ones for grouping; to what extent the relationship to light, growth rate, longevity (ability to hold territory), seed production, seed dispersal distance and dispersal methods have been taken into account in the classification process. This is important because many of these treats were mentioned above in the discussion of ontogeny features, and some will be discussed below.
5. The proposed renewal strategies are strongly related to more traditional life strategies (Grime), which the authors discuss in the other article but do not mention in this one. It seems that a discussion of the links between the proposed classification and the Grime system would make the result clearer. It may also be productive to mention the classifications of trees according to life strategies or population behavior strategies that have been proposed by other authors (Smirnova, 1994; Brzeziecki, 2000; Loehle, 2000 and others). Smirnova and her colleagues closely related population strategies to biological potencies of species realized during ontogeny.
Gap colonisers proposed by the authors are closest to the ruderal (reactive) type of strategy and Gap competitors to the tolerant type, according to Smirnova, 1994. Gap successors and Gap fillers are represented by tree species that are most often attributed to the competitive strategy. Picea abies, Quercus robur and Fraxinus excelsior seem to be the trees with the most poorly understood place in the successional system of hemiboreal forests.
Picea abies is traditionally considered a strong competitor with high shade tolerance and reactive traits. However, difficulties of its regeneration in closed broadleaved forests, including in the hemiboreal zone, have been noted. Currently, in many cases, the main place of successful spruce regeneration on soil is secondary birch and pine forests. This includes forests on abandoned arable land, suggesting that the spread of arable lands was important for the widespread distribution of spruce in the past.
Quercus robur and Fraxinus excelsior are difficult to regenerate in shade forests, which has been controversial for more than a century. The extreme viewpoint is the assumption of a "non-forest" biology of these species and a close connection between their regeneration and grazing by wild ungulates (Frans Vera theory). Smirnova and others (2001*) proposed to identify a special "edge group" of tree species along with early-successional and late-successional species. It includes species whose successful regeneration is associated with meadows and edges after grazing. These include apple, pear, and oak. Currently, the main place of natural regeneration of oak is abandoned meadows and secondary light forests. Of course, even a brief overview of the oak's peculiarities is worthy of a separate article, but it is necessary to note the special position of this species.
The situation with understanding the peculiarities of ash regeneration is even more complicated and it is impossible to summarize it briefly. However, we would like to note that Smirnova et al. (1994) previously considered ash to be capable of sustainable existence in shady broadleaved forests with gap mosaic, including relying on Evstigneev's data on the light-loving nature of ash. However, recent studies show (e.g., Shashkov et al., 2022**) a different result: like oak, ash has no sustainable regeneration in such a forest. At the same time, ash trees successfully spread to abandoned arable land and pastures (European Russian forests, 2017).
6. An important weakness of the classification of Tree regeneration strategies, in my opinion, is the lack of a clear definition of the concept of "gap" (very broad interpretation of this term by the authors). At the same time, the names of species groups are created using this word.
The classical understanding of "gap" in forest science is within the gap-paradigm: a gap in the forest canopy resulting from the death of a tree or group of trees. In the case of species belonging to the Gap coloniser group, it does not refer to gaps themselves, but to glades or other fairly large open spaces. Angelstam and Kuuluvainen (2004) proposed to distinguish three broadly defined types of forest dynamics 1) succession after severe stand-replacing disturbances, 2) cohort dynamics related to partial disturbances and 3) gap dynamics caused by the death of individual trees or small groups of trees. It is desirable that the authors of a peer-reviewed manuscript relate their proposed terms to these, known and fairly widely used terms.
7. I believe that among modern hemiboreal forests it is very rare to find forests close to climax forests with realized gap dynamics. Therefore, additional ecosystem variants should be considered when searching for sites of successful natural regeneration of many tree species. In particular, secondary forests after anthropogenic disturbances or forest plantations provide a number of species with the most stable conditions for regeneration. For example, pine forests (predominantly forest crops) are the site of successful oak regeneration and growth.
Noteworthy are also the vegetation succession systems associated with ungulate grazing and other environment-forming activities of animals. Attention to these cases has grown since Remmert's (1991) mosaic-cyclic concept, the development of disturbance theory, and ideas about keystone species (Mills et al., 1993; Paine, 1995). However, probably revolutionary was the monograph by Vera (2000), after which the introduction of wild herd ungulates or their analogs into natural ecosystems became part of closer-to-nature management.
Perhaps the article by Smirnova, Toropova, 2016 ("Potential ecosystem cover-a new approach to conservation biology") may be of interest when discussing the place of different tree species in successional systems.
The Tree Regeneration Strategies system for tree species in Europe could benefit from greater clarity to account for the diversity of disturbances and forest types. It's important to ensure that the system is easy to understand. The proposed classification's ability to aid forestry practice and management solutions should be better explained, possibly in the discussion section.
8. Line 330-337 the examples given are rather formal and do not reflect the connection of various issues discussed in the "results" section, such as ontogenesis, renewal strategies and others. Earlier widely known publications (e.g., the mentioned article by Angelstam and Kuuluvainen, 2004) are not mentioned. Discussion of this issue seems to be very important and should be improved for the objective of the paper.
It may be fruitful to frame the discussion as an integration of ideas about biological and ecological properties of species into ideas about forest dynamics. Perhaps, taking into account the links "main disturbances - main dynamics variants - species adaptations - management features".
In the Conclusion, the authors say that they have shown specific features of stand dynamics. This is a very important component of the article, but it has not been realized yet. It would be desirable to see a more complete realization of the proposed ideas with examples of taking species biology into account when managing forests with different types of successional dynamics and different stand variants.
9. Line 353-355: speaking about factors affecting natural regeneration, the authors do not mention such an important variable as seed dispersal distance. In many cases, successful spread of species is hindered by short dispersal distances compared to the large size of anthropogenic disturbances. It is likely that the impoverished composition of many modern forests is the result of a long land-use history that has created gaps in the local ranges of tree species. This is probably a factor that needs to be considered and specially addressed by forest management, such as species planting or seed dissemination.
10. Table 5 is barely discussed in the text of the paper. It would have more sense to either discuss how Seed-Bearing Age and Seed Harvest Interval can affect forestry ideas and regulations, or remove that part from the article.
11. Lastly about the title. The title of the article does not seem to be successful and sufficient. "Tree regeneration strategies" is only one of the issues considered. Besides, this part is not very novel: the authors rely on their already published work (Petrokas & Manton, 2023). Probably it would be appropriate to include a reference to "ontogenesis" in the title, as well as to reflect in the title the purpose of the review, which the authors declare as a discussion of the possibilities and ways to integrate the considered concepts to optimize close-to-nature forest management.

*https://www.researchgate.net/publication/346246637_Ocenka_i_prognoz_sukcessionnyh_processov_v_lesnyh_cenozah_na_osnove_demograficeskih_metodov?_sg%5B0%5D=aEVt-ZOzgszjAuPB4lxbdWstniROjjG5ZyzUcYQykdLD5-ROE8mIqXdPpjuqN3y85GcXW5Uk27SwOea-WlJO2ZETa60aQeOlP55hJ6a6QRA.VEGPenp9-gQKp34gUUk8_kDK1-FREtq4yOvnxs2prXZ8PAgh7SALq1Wt4Ljt3Rumg5GvhcawLsS9jxOqmH6LdQ&_tp=eyJjb250ZXh0Ijp7ImZpcnN0UGFnZSI6InB1YmxpY2F0aW9uIiwicGFnZSI6InByb2ZpbGUiLCJwcmV2aW91c1BhZ2UiOiJwcm9maWxlIiwicG9zaXRpb24iOiJwYWdlQ29udGVudCJ9fQ

** https://ncr-journal.bear-land.org/uploads/e7a00ad8f7b1da651e0e462f33682646.pdf

·

Basic reporting

Clear and professional English language used throughout. Intro and background show context. Literature well referenced and relevant.
Structure of the review conforms to PeerJ standards. The review is cross-disciplinary interest and within the scope of the journal. The Introduction adequately introduce the subject of study.

Experimental design

The content of the article corresponds to the goals and topics of the PeerJ .
This review represents a original investigation conducted to the highest technical and ethical standards.
The methods are described in sufficient details. Sources are quoted sufficiently.
The review is organized logically into sequential subsections.

Validity of the findings

This is a very useful review about tree regeneration startegy for for organizing close-to-nature forest management based on undestatding biology and ecology tree species. Conclusions are well stated, linked to original research questions

·

Basic reporting

The article is well-written both linguistically and in terms of content. Sufficient background is provided, complimented by up-to-date and relevant literature references. Introduction introduces the subject adequately and thoroughly, and has a clear topic statement.

Experimental design

It is a literature review article that falls under the fields of biological and environmental sciences, focusing on forest ecology – therefore, the article content is within the aims and scope of the journal.

Validity of the findings

Conclusions are well stated, linked to the research questions and supporting results.

Additional comments

In the hemiboreal vegetation zone, an issue with finding alternatives to clear-cutting forest management practices has been an ongoing discussion, precisely because there is a prevailing opinion that the main shade-tolerant tree species (that is also economically important) is spruce. As spruce is associated with many problems (mainly bark beetle attacks and root rot damages), the implementation of alternative management practices is made difficult. But looking for alternative methods isn't the only approach; improving widely used practices is also possible. This literature review article holds significant importance as it advocates for the adoption of more close-to-nature forest management practices, suggest implementing assisted migration of shade-tolerant species and improve the vitality of different stand forming species.

Comments:
Line 41 – suggest removing “new” from the sentence as close(r)-to-nature forest management approach is not a new one but has gained increasing attention and importance over the past few decades.
Lines 121-135 – recommend referring also to Table 1 in the paragraph.

Additionally, the following citations are missing from the reference list (in order of appearance):
Moser et al., 2020
Claessens et al. (2010)
Harrington et al. (2008)
Navasaitis (2008)
Beck et al. (2016)
Övergaard (2010)
Barbour and Brinkman (2008)
… and references from Table 4 title

·

Basic reporting

English is very good.

Yes, the review is of broad interest within the filed of forest ecology and forest management. This type of review has been done in the past, however, it has been a long time, and there is substantial new material reviewed here, so it seems reasonable to revisit the topic at this time. The literature cited spans over a large time span from decades ago up to the 2023, creating a good time line of development of the science presented.

The introduction makes a good case for how the topic applies to forest management in Europe and how it fills a gap in knowledge and allows an advance in understanding that fulfils forest management goals of the EU and helps foresters to think about how to adapt to climate change.

Ontongeny of tees from seed germination through growth and maturity is a key topic for ecologists and foresters to understand, and therefore good to recognize in forest management, which is one of the reasons for this paper.

Experimental design

The survey methods are very good. It is made clear that this is a review that integrates ideas--not a data analysis or meta analysis or a numerical bibliographic analysis--and it does work as such. Sources are adequately cited and also summarized in the tables and figures in ways that make all of the information that was synthesized, as well as thought processes of the authors, accessible and understandable to the reader. The progression of topics and order in which presented in the manuscript and the four strategies of regeneration that are developed and presented are logical.

Validity of the findings

The actual review results present a unique perspective on development and succession in forests and how these relate to current forest management. Current management generally provides a narrow range of conditions for species with all of the different strategies identified further on in the results section.
The four strategies that can be arranged from large gaps to medium gaps, small gaps and understory growth, with the latter two sometimes also responding by release from suppression makes sense. Table three does a good job of summarizing the growth form attributes of various tree species.
The discussion points out how the four forest dynamic types result in overlapping communities, develop of which is restricted by some forest management strategies. It shows how one species Scots pine has varying dynamics in different site types. Table 4 makes it clear that the same species can very somewhat in regeneration strategy depending on ecosystem type.
These new ideas should allow other scientists to think about them and perhaps respond with similar analyses for forests in other parts of the world.

---

## Round 0.2 · accepted · Accept

Dear Dr. Petrokas,

I am pleased to inform you that your paper has now been accepted for publication in PeerJ.

Sincerely,
Dr. Shaw Badenhorst

·

Basic reporting

The article by Raimundas Petrokas, Michael Manton and Darius Kavaliauskas is devoted to the review of ideas on discrete description of tree ontogeny and tree regeneration strategies for the development of closer-to-nature forest management strategies. This review is of broad interest and within the scope of the journal. The introduction presents in detail and adequately the topic of the review and its relevance for a wide range of specialists related to both forest theory and practice.

Experimental design

The presentation of results and discussion are greatly improved by the authors. Links between different parts of the review have been added. Information on existing concepts has been added and new sources discussed. The quality of the text has been improved so that it can be understood by specialists from different disciplines.

Validity of the findings

The conclusion and abstract are generally consistent with the content of the review, identifying opportunities for future directions in forest management in relation to more fully utilizing knowledge of tree species biology. The authors have improved the linkage of the conclusion to the discussion.

Additional comments

The article may be recommended for publication.